# COVID-19-Related Social Isolation Predispose to Problematic Internet and Online Video Gaming Use in Italy

**DOI:** 10.3390/ijerph19031539

**Published:** 2022-01-29

**Authors:** Umberto Volpe, Laura Orsolini, Virginio Salvi, Umberto Albert, Claudia Carmassi, Giuseppe Carrà, Francesca Cirulli, Bernardo Dell’Osso, Mario Luciano, Giulia Menculini, Maria Giulia Nanni, Maurizio Pompili, Gabriele Sani, Gaia Sampogna, Working Group, Andrea Fiorillo

**Affiliations:** 1Clinical Psychiatry Unit, Department of Clinical Neurosciences, School of Medicine, Università Politecnica delle Marche, Via Tronto 10/A, 60126 Ancona, Italy; laura.orsolini@ospedaliriuniti.marche.it (L.O.); v.salvi@univpm.it (V.S.); 2Department of Medicine, Surgery and Health Sciences, University of Trieste and Department of Mental Health, Azienda Sanitaria Universitaria Giuliano Isontina—ASUGI, 34148 Trieste, Italy; ualbert@units.it; 3Department of Clinical and Experimental Medicine, University of Pisa, 56126 Pisa, Italy; claudia.carmassi@unipi.it; 4Department of Medicine and Surgery, University of Milan Bicocca, 20126 Milan, Italy; giuseppe.carra@unimib.it; 5National Institute of Health, 00161 Rome, Italy; francesca.cirulli@iss.it; 6Department of Biomedical and Clinical Sciences Luigi Sacco, Aldo Ravelli Center for Neurotechnology and Brain Therapeutic, University of Milan, 20157 Milano, Italy; bernardo.dellosso@unimi.it; 7Department of Psychiatry, University of Campania “L. Vanvitelli”, 80138 Naples, Italy; mario.luciano@unicampania.it (M.L.); gaia.sampogna@unicampania.it (G.S.); andrea.fiorillo@unicampania.it (A.F.); 8Department of Psychiatry, University of Perugia, 06132 Perugia, Italy; giuliamenculini@gmail.com; 9Department of Neurosciences and Rehabilitation, University of Ferrara, 44124 Ferrara, Italy; mariagiulia.nanni@unife.it; 10Department of Neurosciences, Mental Health and Sensory Organs, Faculty of Medicine and Psychology, Sapienza University of Rome, 00100 Rome, Italy; maurizio.pompili@uniroma1.it; 11Department of Neuroscience, Section of Psychiatry, University Cattolica del Sacro Cuore, 00100 Rome, Italy; gabriele.sani@unicatt.it; 12Fondazione Policlinico A. Gemelli IRCCS, 00100 Rome, Italy

**Keywords:** COVID-19, gaming disorder, impulsiveness, internet addiction, problematic internet use, smartphone, smartphone addiction

## Abstract

COVID-19 pandemic and its related containment measures have been associated with increased levels of stress, anxiety and depression in the general population. While the use of digital media has been greatly promoted by national governments and international authorities to maintain social contacts and healthy lifestyle behaviors, its increased access may also bear the risk of inappropriate or excessive use of internet-related resources. The present study, part of the COVID Mental hEalth Trial (COMET) study, aims at investigating the possible relationship between social isolation, the use of digital resources and the development of their problematic use. A cross sectional survey was carried out to explore the prevalence of internet addiction, excessive use of social media, problematic video gaming and binge watching, during Italian phase II (May–June 2020) and III (June–September 2020) of the pandemic in 1385 individuals (62.5% female, mean age 32.5 ± 12.9) mainly living in Central Italy (52.4%). Data were stratified according to phase II/III and three groups of Italian regions (northern, central and southern). Compared to the larger COMET study, most participants exhibited significant higher levels of severe-to-extremely-severe depressive symptoms (46.3% vs. 12.4%; *p* < 0.01) and extremely severe anxiety symptoms (77.8% vs. 7.5%; *p* < 0.01). We also observed a rise in problematic internet use and excessive gaming over time. Mediation analyses revealed that COVID-19-related general psychopathology, stress, anxiety, depression and social isolation play a significant role in the emergence of problematic internet use, social media addiction and problematic video gaming. Professional gamers and younger subjects emerged as sub-populations particularly at risk of developing digital addictions. If confirmed in larger and more homogenous samples, our findings may help in shedding light on possible preventive and treatment strategies for digital addictions.

## 1. Introduction

Italy was the first European country to be seriously involved in the first wave of the COVID-19 epidemic starting on 21 February 2020 in the Lombardia region [1,2]. The World Health Organization (WHO) declared the pandemic status on 11 March 2020 [3]. Consequently, governments around the world prompted taking drastic measures aimed at containing the dissemination of the infectious disease, by applying measures including lockdown, self-isolation and placing individuals in home-based quarantine [4,5,6]. Due to the above-mentioned social and mobility limitations, the internet and smartphone devices represented an at hand conduit to be socially connected and to contact other people around the globe [6,7,8]. In particular, social media represent an alternative form of communication for building interpersonal relationships, particularly among youngsters who spend time on the internet to study, play, watch movies, chat, shop and so forth. Moreover, the internet significantly contributed in disseminating information about the COVID-19 pandemic to the general population, representing one of the key factors in fighting the pandemic itself. However, the unmediated online communication channels have also generated an uncontrollable amount of low-quality, biased and false contents, disinformation epidemiology (the so-called “COVID-19 infodemic”) and information overload, which might have in turn predisposed some to the onset of psychological distress symptoms [9,10].

Furthermore, the increasing internet and social media use, particularly through smartphones, may imply the rise of an excessive and uncontrolled use of the internet, smartphones and social media, which may develop into problematic internet use (PIU) or an addiction to it (IA) [11,12,13,14]. IA is defined as a maladaptive pattern of internet usage, leading to clinically significant impairment or psychological distress [11,12,13,14]. Previous studies showed that IA and PIU may be associated with loneliness, social withdrawal, social isolation, emotional instability, depression and high levels of stress, as well as other addictive behaviors [12,13,14,15,16,17]. The reported increase of feelings of social isolation, during the COVID-19 pandemic, may indeed increase the risk of internet-related psychopathology [18,19,20,21,22]. Problematic social media use (PSMU) has been defined as an excessive involvement in social media activities, leading to a disruption of day-to-day activities and interpersonal relationships and the loss of control over social network use, which in turn may lead the subject to develop a strong psychological need to be constantly connected [23]. Furthermore, online gaming and online social networking, besides constituting a mental health issue *per se* whereby they become excessive and uncontrollable in their usage, may indeed represent strong risk factors for further developing a pathological web-based psychopathology, particularly during the COVID-19 situation [24,25,26,27]. In fact, the fear resulting from the COVID-19 situation and the lockdown-related consequences, together with high levels of stress and anxiety, have been mounting [28,29]. A number of factors, including the greater time spent at home due to lockdown, alone or with family members, with high levels of uncertainty and worries regarding the future, financial insecurity and fear related to health consequences, not only contributed to increase stress, anxiety and depression, but also to worsen general psychological wellbeing [28,30,31,32,33,34,35,36,37,38,39,40]. This situation may reinforce behaviors such as PIU, problematic video gaming disorder (PVGD), problematic TV series watching, PSMU, watching online pornography or surfing the internet [15,16,18,20,21,41].

Therefore, we carried out a web-based cross-sectional observational study in order to identify the association between increased levels of stress, anxiety and depression during the current COVID-19 pandemic and its subsequent containment measures, as a mediatory trigger in the emergence and/or maintenance of a problematic use and/or an addiction to the internet, smartphones, online video gaming and TV binge watching, during phases II (May–June 2020) and III (June–September 2020) of the COVID-19 pandemic in the Italian general population [1,2,3].

## 2. Materials and Methods

### 2.1. Study Design

This is a cross-sectional study, part of the COVID Mental hEalth Trial (COMET) study, coordinated by the University of Campania “Luigi Vanvitelli” (Naples), in collaboration with nine Italian collaborating centers (see Acknowledgment section). The Center for Behavioral Sciences and Mental Health of the National Institute of Health in Rome has been involved in the study by supporting the dissemination and implementation of the project according to the clinical guidelines produced by the National Institute of Health for managing the effects of the COVID-19 pandemic. The present study was carried out as a nationwide population-based online survey to evaluate the impact of problematic internet and video games use during the COVID-19 pandemic on the mental health of an Italian sample. The study protocol was approved by the Ethical Review Board of the University of Campania “L. Vanvitelli” (Protocol number: 0007593/i). Further details about the general study design can be found in Giallonardo et al. [29] and in Fiorillo et al. [28].

### 2.2. Survey Sampling

An online survey was set up through EUSurvey, a web platform launched in 2013 by the European Commission. The application, hosted at the Department for digital services (DG DIGIT) of the European Commission, is available to all EU citizens at https://ec.europa.eu/eusurvey (https://ec.europa.eu/eusurvey/runner/COVID19_uso_internet; last accessed on 26 January 2022). The survey took approximately 45 min to be completed. Participants could stop the survey at any time and save their answers as “draft” on the web platform. Furthermore, participants could interact with the principal investigator of the study and with all researchers through email messages at any time during and after study participation. The survey was officially launched on 4 May 2020 (official Italian starting date of Phase II) in the Italian adult population.

A multi-faceted sampling method based on convenience sampling, snowballing and purposive sampling via social media was followed. Information about the study and invitations to take part to the survey, including a hyperlink to the online survey, was regularly posted. The survey was disseminated through a multi-step procedure: (a) email invitation to healthcare professionals and their patients; (b) social media channels (Facebook, Twitter and Instagram); (c) mailing lists of universities, national medical associations and associations of stakeholders (e.g., associations of users/carers); and (d) other official websites (e.g., healthcare or welfare authorities websites).

Inclusion criteria were: 18 years of age or older, understand Italian language and willingness to participate in the study. Eligible participants were requested to complete all sections of the survey, including a socio-demographic section and a battery of self-reported online measures. The study was conducted in accordance with the Declaration of Helsinki (as revised in 2013) [42] and with local regulations.

### 2.3. Clinical Measures

The survey includes a socio-demographic section and a battery of self-reported questionnaires, described below.

The ***Internet Addiction Test (IAT, Italian Version)*** [43] is a 20-item scale measuring the presence and severity of self-reported compulsive use of the internet, including compulsivity, escapism and dependency. The scale categorizes IA as an impulse-control disorder and the term internet refers to all types of online activities. Each item is rated on a 5-point Likert scale from 0 (less extreme behavior) to 5 (most extreme behavior). The IAT total score is the sum of the ratings given for each item. It includes the following subscales: (a) salience (i.e., feelings of worry about the internet, hiding the behavior from others, use of internet as a mental escape from disturbing thoughts; feelings that life without the internet would be boring, empty or joyless); (b) excessive use (i.e., excessive online behavior and compulsive usage); (c) neglect work (i.e., the internet is experienced as a necessary appliance akin to the television, microwave or telephone); (d) anticipation (i.e., the subjects may feel compelled to use the internet when offline); (e) lack of control (i.e., trouble managing his/her online time, frequently stays online longer than intended, others may complain about the amount of time he/she spends online); (f) neglect social life (i.e., online relationships to cope with situational problems and/or to reduce mental tension and stress). The Italian version of the IAT was found to have good psychometric properties, with a good internal consistency (Cronbach’s ***α*** ranged from 0.83 to 0.86) [43]. The severity of PIU and IA, as measured by the IAT, is one of the primary outcomes of the study.

The ***Internet Gaming Disorder Scale (IGDS-short form, Italian version)*** [44] is a 9-item self-reported tool to assess IGD according to the DSM-5 diagnostic criteria [45]. It assesses the severity of PVGD and IGD and its detrimental effects by examining both online and/or offline gaming activities in the previous 12 months. The scale includes 9 items corresponding to the 9 DSM-5 criteria (preoccupation, tolerance, withdrawal, reduce/stop, loss of interests, continued use, deception, escape and conflict) [45]. Each item is ranked on a 5-point Likert scale from 1 (never) to 5 (very often). Higher scores indicate a higher degree of gaming disorder, with a pathological cutoff designed to be below 21 [44]. The Italian version of the IGDS showed excellent reliability with an internal consistency coefficient (Cronbach’s ***α*** = 0.96) [44]. The severity of IGD is one of the primary outcomes of the study.

The ***Depression, Anxiety and Stress Scale (DASS-21, Italian short version)*** [46] is a self-reported questionnaire consisting of 21 items subdivided into three subscales: depression, anxiety and stress. Every item is rated from 0 (did not apply to me at all) to 3 (applied to me very much). The total score assesses the general distress dimension. The DASS-21 is based on a dimensional construct, so that the differences between depression, anxiety and stress experienced by normal subjects and clinical populations are considered essentially as differences of degree. The Italian version of DASS-21 displayed very good reliability, with good or excellent internal consistency values for the total score (Cronbach’s ***α*** ranged from 0.90 to 0.92) and the three scales (Cronbach’s ***α*** ranged from 0.74 to 0.91) [46]. The severity of depressive-anxiety-stress symptoms was considered as a secondary outcome.

The ***Barratt Impulsiveness Scale (BIS-15, Italian version)*** [47] is a short version of the self-reported Barratt Impulsiveness Scale (BIS-11) [48], consisting of 15 items and three subscales. Each item is scored on a 4-point Likert scale from 1 (rarely/never) to 4 (almost always/always). BIS-15 assesses the three main dimensions of impulsive behavior on separate subscales: motor (acting without thinking), attentional (a lack of focus on the ongoing task) and non-planning impulsivity (orientation to the present rather than to the future) [49]. The Italian version of the BIS-15 showed a good internal consistency coefficient (Cronbach’s ***α*** = 0.81) [47]. The severity and type of impulsiveness were considered as secondary outcomes.

The ***Social Adaptation Self-Evaluation Scale (SASS, Italian version)*** [50] is a 21-item scale measuring social motivation and behavior in various areas of social functioning including work and spare time (JT), family and external relationships (FE), intellectual interest (II), social compliance/coping abilities (SC) and control of surroundings (CS). Questions 1 and 2 are mutually exclusive but pooled into a single answer/item (Q1/2, work interest), so that 20 questions are scored giving a total score range of 0 to 60. Each question has four possible responses, from 0 (poorer functioning) to 3 (better functioning). The total score, obtained by the sum of the individual scores, provides a social adjustment index (SAI), so that the higher the value, the better the social adaptation of the individual. The Italian version of SASS demonstrated a good internal consistency coefficient (Cronbach’s ***α*** = 0.752) [50]. A total score above 25 indicates a social maladjustment [50]. The level of social adaptation and functioning due to the COVID-19-related situation was considered as an exploratory outcome.

The ***Aggression Questionnaire (AQ, Italian version)*** [51] is a 29-item self-reported questionnaire designed to measure four major components of aggression (physical aggression, verbal aggression, anger and hostility). Each item is rated on a 5-point Likert scale from 1 (extremely uncharacteristic of me) to 5 (extremely characteristic of me). The questionnaire yields a total score and four scores for primary subscales corresponding to the above-mentioned components of aggression. The Italian version of AQ showed good reliability, with a Cronbach’s ***α*** ranging from 0.85 to 0.83 in a high school student sample while from 0.89 to 0.87 in a university student sample [51]. The severity and type of aggression were considered as secondary outcomes.

The ***Toronto Alexithymia Scale (TAS, Italian version)*** [52] is a 20-item self-reported tool assessing alexithymia (i.e., the difficulty in identifying and describing emotions). It consists of 3 subscales, namely difficulty describing feelings (DDF), difficulty identifying feelings (DIF) and externally oriented thinking (EOT). Items are rated on a 5-point Likert scale from 1 (strongly disagree) to 5 (strongly agree). The total alexithymia score is the sum of all items, whilst the score for each subscale factor is the sum of specific items. The Italian version of the TAS demonstrated a good internal consistency for the TAS total score (Cronbach’s ***α*** = 0.66) and a high test-retest coefficient (r = 0.91) [52]. A cutoff of the TAS total score below 61 is considered pathological [52]. The levels and type of alexithymia were considered as secondary outcomes.

The ***Bergen Social Media Addiction Scale (BSMAS, Italian version)*** [53] is a 6-item self-reported scale on the experiences in the use of social media over a 12-month period. It is an adaptation of the Bergen Facebook Addiction Scale (BFAS) and contains 6 items reflecting the core addiction elements (salience, mood modification, tolerance, withdrawal, conflict and relapse). Each item is rated on a 5-point Likert scale from 1 (very rarely) to 5 (very often) with scores ranging from 6 to 30; a higher score indicates a greater risk of addiction to social media. The Italian version of BSMAS showed good internal consistency (Cronbach’s ***α*** = 0.88) [53]. A cutoff below 16 is considered pathological [53]. The scale is used to assess PSMU.

Finally, a specifically designed ad hoc questionnaire investigating binge watching behavior was administered, consisting of a first screening question asking the respondent whether he/she usually watches TV series, followed by 3 open questions on the intensity and frequency of binge watching and a 7-item questionnaire. Each item of the final questionnaire is ranked on a 4-point Likert scale from 1 (strongly disagree) to 4 (strongly agree), with scores ranging from 7 to 28. A higher score indicates a greater risk of binge-watching behavior.

### 2.4. Statistical Analysis

Descriptive statistics, along with the corresponding 95% confidence interval (CI), were performed in order to describe the socio-demographic and clinical characteristics of the sample. Categorical variables are summarized as frequency (*n*) and percentage (%) whilst continuous variables as means (standard deviation, SD). A chi-square test was used to compare categorical variables.

Student’s *t*-test and two-way tailored analysis of variance (ANOVA) were used to investigate specific dichotomous independent variables (i.e., gender, being infected and/or hospitalized due to COVID-19 infection, being isolated due to COVID-19 infection, being isolated due to a contact with someone affected with COVID-19, having lost job due to the COVID-19 pandemic, having a pre-existing physical disorder, having a pre-existing mental disorder, living alone, professional vs. non-professional gamer, having played video games during the previous 12 months) and their relationship with the severity of PIU and PVGD. In addition, two-way tailored analysis of variance (ANOVA) was used to compare all continuous variables between phase II and phase III of the Italian COVID-19-related pandemic. Findings derived using measures in phase II was considered as indicative of the impact of COVID-19-related lockdown (period March–April 2020), while measures derived by phase III were considered as indicative of the impact of COVID-19-related measures of phase II of the Italian COVID-19 outbreak. Italian regions have been categorized into three subcategories as follows: (a) northern regions (i.e., Friuli Venezia Giulia, Liguria, Lombardia, Piemonte, Valle d’Aosta, Veneto, Toscana and Trentino Alto Adige); (b) central regions (i.e., Marche, Emilia Romagna, Umbria and Lazio); (c) southern regions (i.e., Abruzzo, Basilicata, Calabria, Campania, Molise, Puglia, Sicilia and Sardegna). According to this categorization, these three regional areas were compared for all primary and secondary outcomes using multivariate linear regression models [54].

Multivariate analysis of variance (MANOVA) was used to compare the severity of PIU and PVGD and all other continuous variables considering the fixed factors (i.e., work, educational level, geographic area).

Bivariate Pearson’s correlations were used to investigate potential relationships between primary outcomes (IAT and IGDS-SF total scores) and other secondary variables (Appendix A). Linear regression analysis was performed to investigate the associations between the IAT and IGDS-SF total scores and all other psychopathological measures. The odds ratios (OR), corresponding to 95% of confidence intervals (CI), and standardized coefficient β values were generated for each variable.

Furthermore, considering that DASS total score was established as the primary outcome in the ‘COVID-IT-mental health’ study [28], further analyses were conducted by grouping DASS total score and subscales categories according to the Lovibond and Lovibond [55] severity ratings and comparing our findings with those found in the ‘COVID-IT-mental health’ study [28].

Moreover, PROCESS macro version 3.5.3 (released 11 February 2021) for SPSS [49] was used to carry out moderation (Model 1) and mediation (Model 4) analyses. Mediation analyses were conducted to test whether the direct or indirect effect of BSMAS on IAT or IGDS-SF, IAT on IGDS or BSMAS and IGDS on IAT or BSMAS were mediated by clinical variables which may be directly influenced by COVID-19-related lockdown and outbreak, such as anxiety, depression, stress and/or general psychopathology at DASS-21. The choice of this mediator was also suggested by the previous COVID-IT-mental-health study, whereas DASS-21 was settled as the primary outcome to be investigated as a measure of the impact of the COVID-19 situation on mental health [28]. Indicators of indirect effects were tested using bias-corrected bootstrapping (*n* = 5000) with 95%CI, by setting a statistical significance when the 95%CI does not contain zero. Additional exploratory moderation analyses were conducted to evaluate whether all variables related to COVID-19 (particularly, phase II/phase III, home isolation due to COVID-19 infection or contact with someone affected with COVID-19) might plausibly influence IAT, IGDS and BSMAS total scores.

All analyses were performed using the software Statistical Package for Social Science (SPSS) version 26.0 (IBM SPSS Statistics, Chicago, IL, United States). For all analyses, the level of statistical significance was set at *p* < 0.05, two-tailed.

## 3. Results

### 3.1. Socio-Demographic and Clinical Characteristics

A total of 1385 individuals (62.5% female) mainly living in Central Italy (*n* = 726; 52.4%) voluntarily agreed to complete the survey. The mean age of the sample was 32.5 (SD = 12.9; range = 18–82) years, without statistically significant gender differences (*p* = 0.605). The most frequent educational level is university degree (52.3%; *n* = 724). The total average number of family members (including the respondent) is 4.1 (SD = 1.4) (for further details, see Table 1).

Participants were mainly students (47.7%) or full-time employed (32.7%) (Table 1); 4.1% of respondents declared to have lost their job during the early phases of the COVID-19 pandemic. Overall, 51% of respondents declared to be satisfied (51%, *n* = 706) regarding their current financial situation.

About 89.5% of the sample (*n* = 1239) did not suffer from any pre-existing physical illness before the pandemic; only 6% (*n* = 83) had previously received a diagnosis of a mental disorder before the COVID-19 pandemic (Table 1).

More than 50.8% of respondents (*n* = 704) scored above the threshold of 60 for the DASS-21 total score, indicating a severe general distress/psychopathology. Depressive symptoms were moderate in 43.8% of respondents (*n* = 606). Anxiety symptoms were extremely severe in 77.8% (*n* = 1077). Stress symptoms were mild in most participants (58.3%) (Table 2). Significant higher levels at DASS-21 total score were reported in phase II (60.7, SD = 0.5) compared to phase III (59.0, SD = 0.5) (*p* = 0.019). In addition, significant higher levels of depressive symptoms were reported during phase II compared to phase III (respectively, 20.2, SD = 0.2 and 19.6, SD = 0.2; *p* = 0.033), as well as for stress levels (respectively, 17.2, SD = 0.2 and 16.6, SD = 0.2; *p* = 0.029).

The average SAI score was 49.6 (SD = 5.2), with the majority of the sample lying in the normal functioning range (*n* = 896, 64.7%) (Table 2). Higher SAI scores were found among individuals with pre-existing physical illness (*p* = 0.003) and in those who had isolated due to contact with a subject affected by COVID-19 (*p* = 0.027). SAI negatively correlated with the IAT neglect work subscale (r = −0.066, *p* = 0.013) and BSMAS (r = −0.016, *p* < 0.001). Positive correlations were reported between the SAI and TAS total score (r = 0.106, *p* < 0.001), TAS DDF (r = 0.065, *p* = 0.015), TAS DIF (r = 0.055, *p* = 0.42) and TAS EOT (r = 0.106, *p* < 0.001) (Appendix A).

While 64.1% of the sample (*n* = 888) reported a potential impulse dyscontrol, with a mean score of 39.3 (SD = 7.5) at BIS-15, the average AQ total score was 97.7 (SD = 14.8), with most of the sample being in the range of pathological aggression (*n* = 904, 65.3%), particularly physical aggression (*n* = 1225, 88.5%) and anger (*n* = 934, 67.4%). At the TAS-20 total score, most respondents (*n* = 813, 58.1% of the sample) showed a possible or confirmed alexithymia (Table 2).

### 3.2. Problematic Internet Use

The average IAT score was 46.5 (SD = 10.2), without any statistically significant gender difference (*p* = 0.719). PIU, as measured as a moderate level of internet usage on the IAT scale, was found in 33.1% of the sample. Severe levels of IA were present in 0.8% of respondents (Table 2). Significantly higher IAT total scores were found amongst single respondents (55.8, SD = 2.7) compared to married and separated/divorced subjects (*p* < 0.005). Moreover, significantly higher IAT total scores were reported amongst participants infected with COVID-19 and those quarantined due to contact with an individual affected with COVID-19 infection (56.0, SD = 11.3) (*p* = 0.006), even though the most significant association appeared to be related to home isolation (*p* < 0.005). However, a statistically significant difference between students who had been isolated due to contact with someone who had been infected with COVID-19 was reported for the IAT “Lack of Control” subscale (*p* = 0.040). During phase III participants reported statistically significant higher levels for IAT neglect work (6.6, SD = 0.1), compared to those in phase II (6.6, SD = 0.1; *p* = 0.008). Moreover, they showed higher IAT total scores (46.9, SD = 0.4), when compared to phase II (45.9, SD = 0.4), as well as higher IAT anticipation scores (respectively, 4.8, SD = 0.1; and 4.6, SD = 0.1), although just reaching a trend value for statistical significance (respectively, *p* = 0.084 for IAT total score and *p* = 0.058 for IAT anticipation subscale).

#### 3.2.1. Problematic Internet Use and Videogaming

Participants who defined themselves as “professional gamers” and those who played video games during the previous 12 months reported significantly higher IAT total scores (55.7, SD = 13.0), when compared to nonprofessional gamers (*p* = 0.048). Mediation analyses showed that IAT significantly predicts IGDS and that their interaction was significantly mediated by DASS depressive symptoms (ß = +0.0113, 95%CI (0.0012–0.0221), *p* < 0.001) (Figure 1).

#### 3.2.2. Problematic Internet Use and Problematic Social Media Use

Positive correlations were found between the IAT total score and the BSMAS (r = 0.465, *p* = 0.01). Mediation analyses showed that the IAT significantly predicted the BSMAS and that their interaction was significantly mediated by general psychopathology, as measured using the DASS total score (ß = +0.0522, 95%CI (0.0401–0.0655)), depressive symptoms at the DASS depressive subscale (ß = +0.0420, 95%CI (0.0308–0.0541)), anxiety symptoms at the anxiety subscale (ß = +0.0298, 95%CI (0.0204–0.0400)) and stress levels at the DASS stress subscale (ß = +0.0334, 95%CI 90.0239–0.0437), *p* < 0.001) (Figure 2).

### 3.3. Problematic Internet Gaming

The mean score of total IGDS-SF was 13.1 (SD = 6.3), without any statistically significant gender difference (*p* = 0.599). Based on total IGDS-SF scores, 13.8% of participants were classified as having a problematic level of PVGD, with an average of 4.2 (SD = 9.8) hours per week spent on video gaming. Higher levels of IGDS total score were associated with being a student and being single, but not with educational level and age. No statistically significant differences were found in IGDS total scores among participants with a physical illness (*p* = 0.213), whereas significant differences at IGDS-SF total scores were observed among subjects with previous mental illness (*p* = 0.039).

No significant differences in IGDS total scores were reported amongst participants who had versus had not lost their job due to COVID-19 (*p* = 0.675). No statistically significant differences in IGDS total scores were found in infected/hospitalized or quarantined subjects, while a statistically significant difference was observed for the IGDS-SF total scores among participants isolated due to contact with a COVID-19+ subject (*p* = 0.005).

#### 3.3.1. Problematic Internet Gaming Disorder and Playing Videogames

Overall, participants who declared that they had played video games during the previous 12 months were mostly females (62.5%) and students (69.4%, *p* < 0.005), and they reported higher levels of IGDS-SF, compared to those who did not declare that they had been video gaming during the previous year (*p* < 0.005). Those respondents also reported statistically significant higher BIS total scores (*p* = 0.005), BIS attentional subscale (*p* = 0.023), AQ total scores (*p* < 0.005), AQ physical aggression subscale (*p* < 0.005), AQ verbal aggression subscale (*p* = 0.037), AQ anger subscale (*p* = 0.038), AQ hostility subscale (*p* < 0.005), TAS DIF subscale (*p* = 0.004), IAT total score (*p* < 0.005), IAT salience subscale (*p* < 0.005), IAT excessive use subscale (*p* < 0.005), IAT neglect work subscale (*p* = 0.005), IAT lack of control subscale (*p* = 0.002), IAT neglect social life subscale (*p* < 0.005), IGDS total score (*p* < 0.005), Bergen social network total score (*p* = 0.001) and binge watching total score (*p* < 0.005). The minority of subjects (2.5%; *n* = 34), who defined themselves as “professional gamers” (64.7% of them were students), also displayed a statistically significant higher IGDS total score (*p* = 0.015).

#### 3.3.2. Problematic Internet Gaming Disorder and Problematic Internet Use

Mediation analyses showed that the IGDS significantly predicted the IAT and that their interaction is significantly mediated by general psychopathology, as measured via DASS total score (ß = +0.0830, 95%CI (0.0584–0.1095), *p* < 0.001), depressive symptoms (ß = +0.0838, 95%CI (0.0591–0.1113), *p* < 0.001), anxiety symptoms (ß = +0.0547, 95%CI (0.0343–0.0784), *p* < 0.001) and stress levels at DASS (ß = +0.0366, 95%CI (0.0198–0.0546), *p* < 0.001) (Figure 3).

#### 3.3.3. Problematic Internet Gaming Disorder and Problematic Social Media Use

A significant positive correlation was found between IGDS total scores and BSMAS total scores (*p* < 0.005). Mediation analyses showed that the IGDS significantly predicted the BSMAS and that their interaction was significantly mediated by general psychopathology, as measured via the DASS total score (ß = +0.0677, 95%CI (0.0480–0.0883), *p* < 0.001), depressive symptoms (ß = +0.0635, 95%CI (0.0458–0.0826), *p* < 0.001), anxiety symptoms (ß = +0.0432, 95%CI (0.0269–0.0608), *p* < 0.001) and stress levels on the DASS (ß = +0.0335, 95%CI (0.0183–0.0497), *p* < 0.001) (Figure 4).

### 3.4. Problematic Social Media Use

Around 77.8% of the participants were classified as PSMU users, with a mean total BSMAS score of 20.5 (SD = 6.6), without significant gender differences (*p* = 0.620). The use of social media was not greater in subjects with mental illness (*p* = 0.676) compared to the general sample, while subjects suffering from a physical illness showed higher BSMAS total scores (although just reaching a trend value for statistical significance, *p* = 0.068). Moreover, 87.2% of students displayed problematic social network use.

Significant higher BSMAS total scores were reported amongst those participants who declared they had lost their job due to the COVID-19-related situation compared to those who maintained their job (*p* = 0.043). No statistically significant differences in BSMAS total scores were found among those who had been infected with COVID-19 or hospitalized due to a COVID-19 infection and those who had been quarantined or isolated due to COVID-19. Moderation analysis showed that the BSMAS is a significant predictor of the IAT (F_3,1381_ = 131,08, *p* < 0.01, R^2^ = 0.22) and that this interaction is significantly moderated by the condition of home isolation due to contact with someone affected with COVID-19 (*p* = 0.0436, 95%CI ((−0.07411)–(−0.108)) (Figure 5).

Moderation analysis showed that the BSMAS is a significant predictor of the IGDS (F_3,1381_ = 21,32, *p* < 0.01, R^2^ = 0.04) and that this interaction is significantly moderated by the COVID-19 phase in both phases (Phase II: *p* < 0.01, 95%CI (0.1771–0.3223); Phase III: *p* < 0.01, 95%CI (0.0791–0.2127)), with a more pronounced effect in phase II (b = +0.2497) compared to phase III (b = +0.1459) (Figure 6).

#### Problematic Social Media Use, Internet and Videogaming

No statistically significant differences were found for the BSMAS total score amongst professional gamers or among those who declared they had played video games during the previous 12 months. Mediation analyses showed that the BSMAS significantly predicts the IAT and that their interaction is significantly mediated by general psychopathology, as measured via the DASS total score (ß = +0.1152, 95%CI (0.0843–0.1483), *p* < 0.001), depressive symptoms (ß = +0.1047, 95%CI (0.0778–0.1344), *p* < 0.001), anxiety symptoms (ß = +0.0694, 95%CI (0.0451–0.0963), *p* < 0.001) and stress levels (ß = +0.0574, 95%CI (0.0340–0.0815), *p* < 0.001) (Figure 7).

Furthermore, mediation analyses also revealed that the BSMAS significantly predicts the IGDS and that their interaction is significantly mediated by general psychopathology, as measured via the DASS total score (ß = +0.0463, 95%CI (0.0262–0.0683), *p* < 0.001), depressive symptoms (ß = +0.0474, 95%CI (0.0291–0.0668), *p* < 0.001), anxiety symptoms (ß = +0.0301, 95%CI (0.0146–0.0469), *p* < 0.001) and stress levels (ß = +0.0153, 95%CI (0.0011–0.0313), *p* < 0.001) (Figure 8).

### 3.5. Binge Watching

The mean score for binge watching was 20.2 (SD = 9.2), without any statistically significant gender differences (*p* = 0.513). No significant differences were observed in binge watching between those who have a physical (*p* = 0.122) or a mental (*p* = 0.772) illness. No statistically significant differences in binge watching total scores were reported among those who had lost their job due to the COVID-19 situation, compared to those who maintained their job (*p* = 0.139), neither among those who had been infected with COVID-19 or hospitalized due to a COVID-19 infection and those who had been quarantined or isolated due to COVID-19. A statistically significant difference was found among those who had played video games during the previous 12 months (*p* < 0.005), while no significant differences were found for BSMAS total score amongst professional gamers compared to the general sample.

## 4. Discussion

The present study evaluated the impact of COVID-19 and its related containment measures on internet usage, online video gaming, social media use and binge watching in the general population, during phases II and III of the COVID-19 outbreak in Italy. Specifically, we hypothesized an increasing prevalence trend in problematic/addicted internet, online gaming and social media use and binge watching along the two phases of the pandemic. Furthermore, our hypotheses were mainly addressed at investigating these variables as potential moderators/mediators of mental health in the general population, by evaluating whether internet use during the COVID-19 pandemic may act as a protective or a risk factor for mental health in the general population.

Overall, a general distress/psychopathology was reported in half of the sample (50.8%), with a severe-to-extremely severe depressive symptomatology in 46.3% of the sample and an extremely severe anxiety symptomatology in 77.8% of the sample (Table 2). Compared to that of the COMET study [27], our sample reported significant higher levels of severe-to-extremely-severe depressive symptoms (46.3% vs. 12.4%; *p* < 0.01) and extremely severe anxiety symptoms (77.8% vs. 7.5%; *p* < 0.01). However, most participants (64.7%) reported normal functioning on a social adjustment index, with no participant reporting social maladjustment. No associations were found between the SAI and our primary outcomes, while negative correlations were found with the IAT neglect work subscale and BSMAS scores, despite previous studies reporting significant correlations between poor social adaptation on the SASS and worry about a gaming disorder, loss of interest in other activities and loss of significant relationships [50,56,57].

Furthermore, our findings indicate PIU (as measured via the IAT) in 33.1% of the participants and in 40.7% of students (with 0.8% of the total sample having developed severe IA, in comparison to 1.2% of young participants having developed severe IA). However, when comparing our data with previous published studies conducted during the COVID-19 pandemic in other countries, our percentages are lower than those measured by other international studies [18,58,59]. For example, a recent study reported the presence of PIU in 46.8% of the sample, while in a Chinese sample IA was observed in 2.7% of the sample and PIU in 33.4% of the cases [58]. A study from Indonesia reported a 14.4% prevalence of IA during the COVID-19 pandemic, with significantly higher risk reported amongst those having COVID-19 confirmed/suspected within the household [59]. However, our findings appear consistent with those reported in previous Italian studies (not conducted during the COVID-19 pandemic) but which recruited only young people and/or university students [60,61,62]. A possible reason for this may lie in the characteristics of our sample, mainly represented by young people (35.4%), young adults (25–29 years; 22.1%) and students (47.7%), which may limit the generalizability of our findings to the general population. In fact, it has been well documented that younger people are at greater risk of the negative effects of out-of-control internet use, due to a hypoactive reflective system and an overactive impulsive system to manage their internal states [63,64,65]. Moreover, a dysregulation of the interoceptive awareness system may increase the incentive salience of internet use as well as the feeling of craving deriving from its compulsive use [64,66]. In this regard, our findings reported significant positive correlations between the IAT total score and the BIS attentional and motor subscales, as previously reported in other published studies [67,68,69]. Moreover, a positive correlation was found between IAT and AQ physical aggression, TAS total score and all its subscales, as documented in previous studies [21,54]. In addition, a positive correlation was found between the IAT, IGDS and BSMS, by underlining the potential mediatory role of the internet in incenting online addictive behaviors such as PVGD and social media addiction [70]. Moreover, a statistically significant mediatory role of COVID-19-related depressive symptomatology was reported in mediating the interaction between the IAT and IGDS. These findings may underline how current COVID-19-induced depressive symptomatology may act as a mediatory trigger in the development and maintenance of behavioral addictions, as already demonstrated in previous studies [37,38,71,72,73]. Moreover, our findings reported a prevalence which is relatively lower than that reported in the early COVID-19 phases in Italy, with an increasing trend for PIU and IA from phase II towards phase III of the COVID-19 outbreak. This increasing trend over time may be more likely related to the cumulative effect of internet usage during the previous phase I and phase II.

Regarding the PVGD, our sample seems likely representative and therefore useful in exploring the impact of the COVID-19 outbreak on PVGD, according to the ICD-11 definition [58,74]. Our findings documented that smartphones and tablets represent the primary electronic devices for accessing the internet and video gaming, followed by PCs/laptops [37,71,75,76,77]. The smartphone is the most frequently used digital device in the young population, as documented in previous studies which reported that around 80% of Italian young people use tablets or smartphones (compared to the prevalence in the general population who reported a rate of 0.8%) [78]. Therefore, these findings may be better understood when we consider that our sample is mainly represented by youngsters. Moreover, our findings did not allow a classification of video gamers to be provided according to Lee et al. [79]. However, we documented positive correlations between the IGDS and BIS attentional and motor subscales and between the IGDS and TAS total score and its subscales (DDF and EOT), leading us to suppose that our game players may more likely belong to the impulsive/aggressive players category (category 1) [77]. In this regard, previous studies reported that individuals with high levels of impulsivity use online games to escape from reality and avoid negative emotions deriving from misrepresented beliefs regarding themselves, others and the surrounding world [80,81]. The IGDS may work as a mediating factor on loneliness and four components of aggression on the AQ [59,82]. Moreover, those video gamers who played during the previous 12 months are those who reported statistically significant higher IAT, IGDS, BSMAS and binge watching scores, compared to those who declared not having played during the previous 12 months.

Finally, our sample appears to be more likely represented by individuals with a problematic use of social media (77.8% of the participants), with a trend of higher BSMAS total scores found among individuals affected with a physical illness. Moreover, significant higher BSMAS total scores were observed among those participants who declared they had lost their job due to COVID-19 (*p* = 0.043) and those coming from Northern Italy (*p* = 0.001). These findings may potentially explain the higher levels of anxiety and depressive symptomatology of our sample, as social media use influences subjective perceived threat of COVID-19, which may indirectly increase anxiety and depressive levels [83,84] and may act as indicator of loneliness in the youngest during the COVID-19 outbreak [85,86,87].

However, our findings should be interpreted in the light of several limitations. First, the purpose of the survey was to examine the effect of the COVID-19 pandemic and relative lockdown situations on PIU and PVGD, by stratifying the findings according to phase II and phase III of the COVID-19 Italian lockdown and according to the geographical region of participants. However, as the survey was not run out during phase I of the COVID-19 Italian lockdown, our findings are not able to show the baseline level of PIU and PVGD in the early stages of the COVID-19 outbreak and no comparative analyses can be made between the different phases. Moreover, geographical stratification revealed that our sample is more represented by individuals coming from Central Italy. For this reason, our findings may not be fully generalizable to the northern regions, which were more deeply affected by the COVID-19 pandemic. Despite these limitations, we found significant differences and increasing trends in PIU, IA, PVGD and social media use, independently from the geographic region of participants.

Secondly, our sample appears to be overrepresented by students and young people, which may explain the high levels of PIU, IA, PVGD and social media use, but whether these findings are generalizable to Italian general population is not adequately investigated. Moreover, the number of subjects who declared they were professional gamers is 2.5%, hence, our survey may not be representative of professional video gamers. In addition, we could not reach people without an internet connection.

Thirdly, our web-based online cross-sectional study design may have not allowed an evaluation of changes in the levels of severity of PIU, IA, PVGD and social media use over time. However, the comparison between primary and secondary outcomes between phase II and phase III helped us in better understanding the occurring trend in PIU, IA, PVGD and social media use over the course of the COVID-19 outbreak.

In addition, the use of self-reported measures, rather than more objective clinician-based assessments, may limit the generalizability of our findings. Consequently, there may be methodological biases when participants answer questions (e.g., social desirability, casual answers due to distraction or the lack of a face-to-face interviewer). Moreover, data were obtained from the respondents’ declarations, by potentially determining inference errors if a generalization process is carried out to the general population. Furthermore, self-report questionnaires do not assess other potential addictive behaviors such as cigarette smoking, alcohol drinking and other voluptuary behaviors. Moreover, our questionnaire did not assess premorbid temperamental and/or personality features, comorbid psychiatric diseases and/or sleep disorders of participants. In addition, the online survey design study may cause response biases due to non-sampling (cfr. snowball sampling), and the recruitment methods may limit the generalizability of the present findings, being mainly based on snowball sampling, which could lead to selection biases. Another limitation regards the use of a non-standardized tool to assess binge watching behavior, even though it was ad hoc developed.

Finally, our survey did not explore other variables, such as main motivations for using the internet (i.e., academic/occupation-related, social media, seeking information, entertainment, etc.). Moreover, the survey did not explore online versus offline video gaming, the type of video game, the motivations of video gaming (i.e., achievement, socializing, escapism, etc.) or other personality and temperamental features which may allow us to clearly classify our video gamers.

Despite the above-mentioned limitations, we collected a large sample, being constituted mainly of young people, so it could be helpful in deeper understanding and better investigating the impact of the COVID-19-related situation with regards the development and maintenance of behavioral addictions in the younger generation, which has often been identified as one of the sub-populations most sensitive to COVID-19-related risks. Moreover, as our sample is overrepresented by female participants, it could be helpful to better characterize gender-based differences in the development and maintenance of behavioral addictions, even though stratified statistical analyses did not report clinically relevant differences. In addition, the use of standardized, validated and widely administered instruments gave the possibility to compare our findings with those already published referring both to COVID-19-related and nonCOVID-19-related situations. Finally, our sample also allowed us to investigate the state-of-the-art of the current impact of lockdown and related restrictions on a large sample of non-clinical individuals.

Obviously, the present findings should be replicated by increasing the sample size by using a longitudinal prospective study design that may provide insights into the causality of the significant associations found amongst the variables here examined. Moreover, a further survey may be proposed in order to compare our findings and those collected after one year, in order to evaluate trends (if any) in PIU, PVGD and IA over time. Finally, it could be interesting to perform a study specifically addressing young people and university students with a more punctual stratification of the sample as well as build a protocol to investigate how the trend in PIU, PVGD and IA changed in professional gamers and/or in those subjects who were already pathological gamers, during the current COVID-19-related condition. Moreover, a further study could evaluate the impact of different affective temperaments, attachment styles and emotional dysregulation attitudes on the trend of PIU, PVGD and IA, as already preliminarily reported in previous studies focusing on the general psychological impact of the COVID-19 outbreak [28,81]. Furthermore, more studies could also investigate the impact of this increasing trend of PIU, PVGD and IA in facilitating the de novo occurrence and/or maintenance of a Hikikomori-like condition following the COVID-19 pandemic [88,89,90].

## 5. Conclusions

Overall, our findings may have significant implications for research and intervention, by providing substantial evidence of excessive internet and video game use during the COVID-19 pandemic. Although our findings do not provide an overview of phase I, interesting considerations may be derived by findings coming from phase II and phase III of the COVID-19 pandemic that may direct preventive and treatment strategies also for the subsequent phase IV. During the first wave of the COVID-19 pandemic, an abrupt and dramatic change in mental health services was reported in Europe [91]. For instance, some mental health services and infrastructures drastically reduced the mental health access and treatment, due to the COVID-19-related restrictions, while other mental health services limited the access only to severe and persistent mental health illnesses. Simultaneously, an increasing trend of de novo occurrence of mental health conditions was observed due to the COVID-19-related condition. In this regard, the observed increasing trend in behavioral addictions from phase II to phase III, also in those individuals without a previous mental health disorder, could represent a cause of clinical concern for mental health professionals. Therefore, implementing preventive strategies (screening of at-risk vulnerable individuals and the general population, education about the responsible use of digital devices and prevention of overflow of digitally-driven information), early identification of problematic use of the internet, social media, video gaming and technological devices as well as prompting therapeutic approaches may address future national plans. Moreover, mental health and addiction services should adequately intercept the at-risk vulnerable population and treat those problematic behavioral conditions which may develop into a behavioral addiction, particularly during pandemic situations like that determined by COVID-19 infection [91,92,93,94,95,96]. In particular, looking at this trend over time should direct mental health professionals towards preventive and treatment strategies to be integrated within the provisional planning for mental health post-COVID-19 [97,98,99,100,101].

Finally, our study might also be useful in informing mental health professionals and researchers towards further research plans. Although our sample might not be fully representative of the Italian population (being mainly overrepresented by females, university students and young people), further research studies should also consider larger sample analysis considering gender-, age- and generation-based stratification. Of course, generalizing research findings into policy is a particularly sensitive issue and it requires extensive research and particular caution [102]. In this regard, a post-hoc analysis specifically selecting this young subsample could be helpful to analyze how the variable age may influence our findings. Moreover, our research should be replicated at international level by comparing country-based findings and evaluating whether our findings should be limited only to the Italian population or rather could represent a first preliminary research direction which should be furtherly investigated. In addition, as already stated, our preliminary findings should be replicated in a larger longitudinal prospective study design to investigate the causality of the significant associations found amongst the variables here examined as well as to evaluate chronological trends (if any) in PIU, PVGD and IA over time across different countries worldwide.

## Figures and Tables

**Figure 1 ijerph-19-01539-f001:**
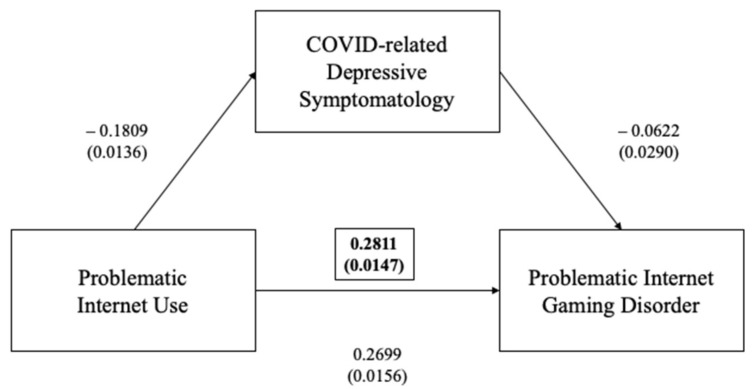
Mediation effect of DASS depression subscale between problematic internet use (as measured via IAT) and problematic internet gaming disorder (as measured via IGDS-SF) (in brackets, total effect).

**Figure 2 ijerph-19-01539-f002:**
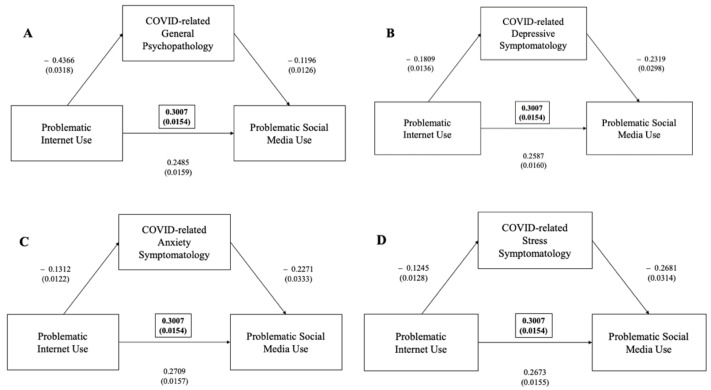
(**A**) Mediation effect of DASS general psychopathology scale between problematic internet use (as measured via IAT) and problematic social media use (as measured via BSMAS) (in brackets, total effect); (**B**) mediation effect of DASS depressive subscale between problematic internet use (as measured via IAT) and problematic social media use (as measured via BSMAS) (in brackets, total effect); (**C**) mediation effect of DASS anxiety subscale between problematic internet use (as measured via IAT) and problematic social media use (as measured via BSMAS) (in brackets, total effect); (**D**) mediation effect of DASS stress subscale between problematic internet use (as measured via IAT) and problematic social media use (as measured via BSMAS) (in brackets, total effect).

**Figure 3 ijerph-19-01539-f003:**
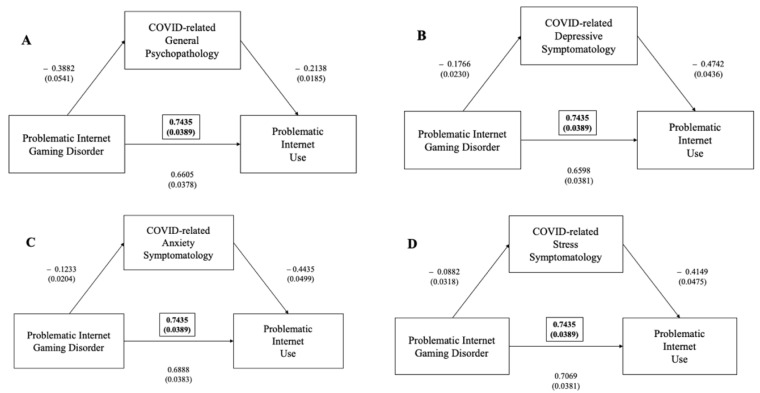
(**A**) Mediation effect of DASS general psychopathology scale between problematic gaming disorder (as measured via IGDS-SF) and problematic internet use (as measured via IAT) (in brackets, total effect); (**B**) mediation effect of DASS depressive subscale between problematic gaming disorder (as measured via IGDS-SF) and problematic internet use (as measured via IAT) (in brackets, total effect); (**C**) mediation effect of DASS anxiety subscale between problematic gaming disorder (as measured via IGDS-SF) and problematic internet use (as measured via IAT) (in brackets, total effect); (**D**) mediation effect of DASS stress subscale between problematic gaming disorder (as measured via IGDS-SF) and Problematic internet Use (as measured via IAT) (in brackets, total effect).

**Figure 4 ijerph-19-01539-f004:**
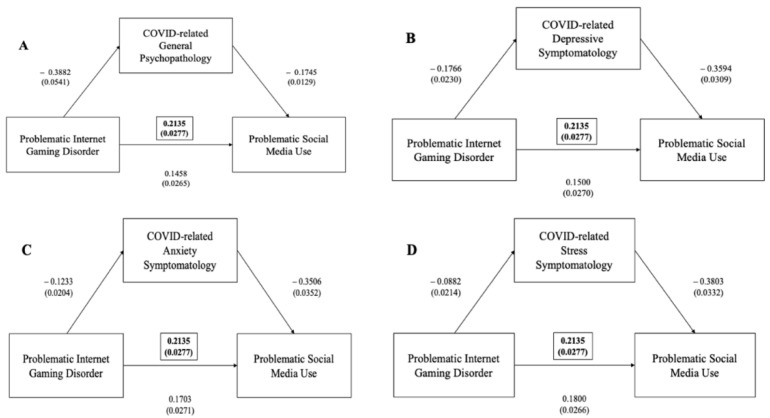
(**A**) Mediation effect of DASS general psychopathology scale between problematic internet gaming disorder (as measured via IGDS-SF) and problematic social media use (as measured via BSMAS) (in brackets, total effect); (**B**) mediation effect of DASS depressive subscale between problematic internet gaming disorder (as measured by IGDS-SF) and problematic social media use (as measured via BSMAS) (in brackets, total effect); (**C**) mediation effect of DASS anxiety subscale between problematic internet gaming disorder (as measured via IGDS-SF) and problematic social media use (as measured via BSMAS) (in brackets, total effect); (**D**) mediation effect of DASS stress subscale between problematic internet gaming disorder (as measured via IGDS-SF) and problematic social media use (as measured via BSMAS) (in brackets, total effect).

**Figure 5 ijerph-19-01539-f005:**
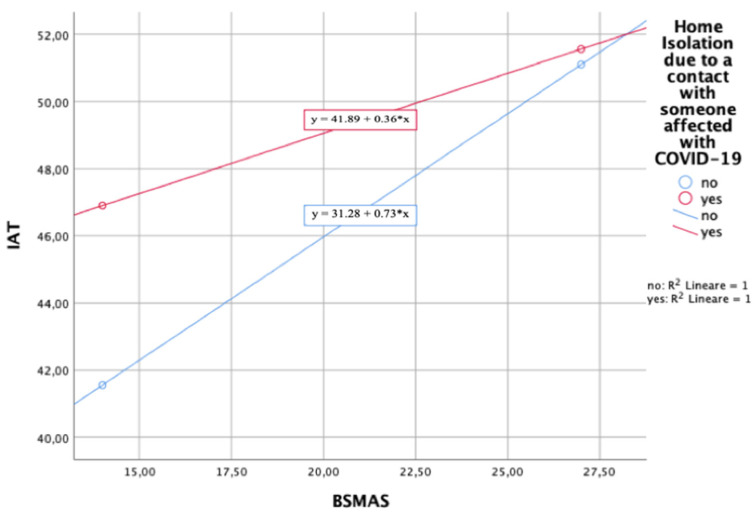
Plot of the interaction between BSMAS and IAT (moderator = isolation due to a contact with someone affected with COVID-19).

**Figure 6 ijerph-19-01539-f006:**
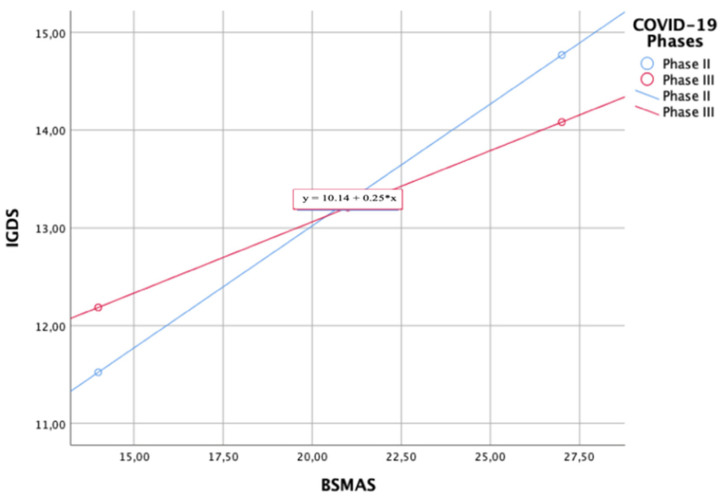
Plot of the interaction between BSMAS and IGDS (moderator = COVID-19 phase).

**Figure 7 ijerph-19-01539-f007:**
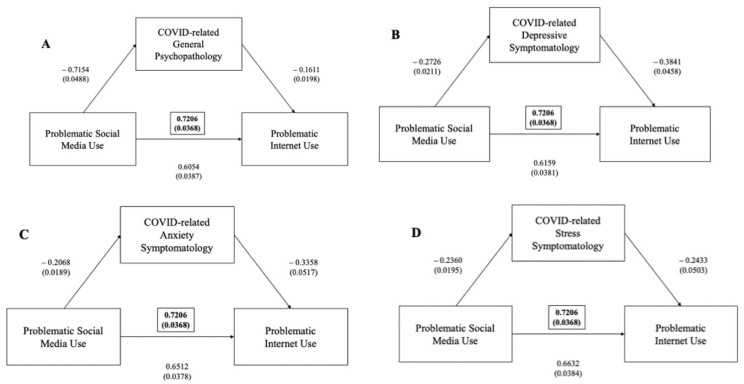
(**A**) Mediation effect of DASS general psychopathology scale between problematic social media use (as measured via BSMAS) and problematic internet use (as measured via IAT) (in brackets, total effect); (**B**) mediation effect of DASS depressive subscale between problematic social media use (as measured via BSMAS) and problematic internet use (as measured via IAT) (in brackets, total effect); (**C**) mediation effect of DASS anxiety subscale between problematic social media use (as measured via BSMAS) and problematic internet use (as measured via IAT) (in brackets, total effect); (**D**) mediation effect of DASS stress subscale between problematic social media use (as measured via BSMAS) and problematic internet use (as measured via IAT) (in brackets, total effect).

**Figure 8 ijerph-19-01539-f008:**
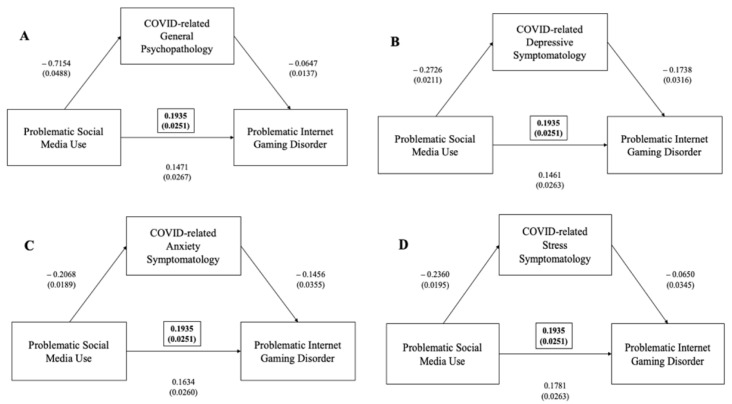
(**A**) Mediation effect of DASS general psychopathology scale between problematic social media use (as measured via BSMAS) and problematic internet gaming disorder (as measured via IGDS-SF) (in brackets, total effect); (**B**) mediation effect of DASS depressive subscale between problematic social media use (as measured via BSMAS) and problematic internet gaming disorder (as measured via IGDS-SF) (in brackets, total effect); (**C**) mediation effect of DASS anxiety subscale between problematic social media use (as measured via BSMAS) and problematic internet gaming disorder (as measured via IGDS-SF) (in brackets, total effect); (**D**) mediation effect of DASS stress subscale between problematic social media use (as measured via BSMAS) and problematic internet gaming disorder (as measured via IGDS-SF) (in brackets, total effect).

**Table 1 ijerph-19-01539-t001:** Socio-demographic characteristics of the sample (*n* = 1385).

**Age, Years, Mean ± SD**	32.5 ± 12.9
18–19 years old, % (*n*)	3.7% (52)
20–24 years old, % (*n*)	31.7% (439)
25–29 years old, % (*n*)	22.1% (306)
30–39 years old, % (*n*)	18.5% (256)
≥40 years old, % (*n*)	24.0% (332)
**Gender, % (*n*)**	
Female	62.5% (865)
Male	37.5% (520)
**Marital status, % (*n*)**	
Single	59.7% (827)
Married or cohabiting	36.4% (504)
Separated or divorced	3.3% (46)
Widowed	0.6% (8)
**Education level, % (*n*)**	
University degree, % (*n*)	52.3% (724)
High school degree, % (*n*)	46.6% (645)
Middle school, % (*n*)	1.1% (15)
Elementary school, % (*n*)	0.1% (1)
**Employment level, % (*n*)**	
Full-time employed, % (*n*)	32.7% (453)
Unemployed, % (*n*)	3.6% (50)
Student, % (*n*)	47.7% (661)
Full-time homemaker, % (*n*)	7.7% (106)
**Lost job due to the pandemic, % (*n*)**	4.1% (57)
**Any comorbid physical condition(s), % (*n*)**	10.5% (146)
**Any mental health problem(s), % (*n*)**	6% (83)
**Have you been infected by COVID-19, % (*n*)**	1.4% (19)
**Have you been isolated due to COVID-19 infection, % (*n*)**	1.9% (27)
**Have you been hospitalized due to COVID-19 infection, % (*n*)**	0.1% (2)
**Have you been isolated due to a contact with someone affected by COVID-19, % (*n*)**	4% (56)
**Have you played any video games in the last 12 months, % (*n*)**	60.4% (836)
**Are you a professional video gamer, % (*n*)**	2.5% (34)
**Hours per week in video gaming, mean ± SD**	4.2 ± 9.8
**Device used in video gaming, % (*n*)**	
Computer, % (*n*)	26.4% (366)
Smartphone, % (*n*)	58.9% (816)
TV, % (*n*)	14.7% (203)

*n*: frequency; %: percentage; SD: standard deviation.

**Table 2 ijerph-19-01539-t002:** Clinical characteristics of the sample (*n* = 1385).

**IAT Total Score, Mean ± SD (Range: 0–100)**	46.5 ± 10.2
Normal level of internet usage (range: 0–30), *n* (%)	23 (1.6%)
Mild level of internet usage (range: 31–49), *n* (%)	893 (64.5%)
Moderate level of internet usage (range: 50–79), *n* (%)	458 (33.1%)
Severe level of internet usage (range: 80–100), *n* (%)	11 (0.8%)
*IAT, Salience subscale, mean ± SD*	8.9 ± 3.5
*IAT, Excessive Use subscale, mean ± SD*	11.1 ± 3.8
*IAT, Neglect Work subscale, mean ± SD*	6.4 ± 2.9
*IAT, Anticipation subscale, mean ± SD*	4.7 ± 1.9
*IAT, Lack of Control subscale, mean ± SD*	11.8 ± 2.6
*IAT, Neglect Social Life subscale, mean ± SD*	3.5 ± 1.6
**IGD9-SF total score, mean ± SD (range: 9–45)**	13.1 ± 6.3
Normal level (range: 9–20), *n* (%)	1194 (86.2%)
Pathological level (range: ≥21), *n* (%)	191 (13.8%)
** *DASS-21, Stress subscale, mean ± SD* **	16.8 ± 5.0
Normal (range: 0–10), *n* (%)	115 (8.3%)
Mild (range: 11–18), *n* (%)	807 (58.3%)
Moderate (range: 19–26), *n* (%)	353 (25.5%)
Severe (range 27–34), *n* (%)	110 (7.9%)
Extremely severe (range: 35–42), *n* (%)	0 (0%)
**BIS-15 total score, mean ± SD (range: 15–60)**	39.3 ± 7.5
*BIS-15, Attentional Impulsiveness subscale, mean ± SD*	11.2 ± 3.5
*BIS-15, Motor Impulsiveness subscale, mean ± SD*	12.7 ± 4.5
*BIS-15, Nonplanning Impulsiveness subscale, mean ± SD*	10.7 ± 2.4
**Social Adjustment Index (SAI), SASS total score, mean ± SD (range: 0–60)**	49.6 ± 5.2
Normal (range: 35–52), *n* (%)	896 (64.7%)
Better functioning (range: ≥52), *n* (%)	489 (35.3%)
Social maladjustment (range: <25), *n* (%)	0 (0%)
**AQ total score, mean ± SD (range: 29–145)**	97.7 ± 14.8
*AQ, Physical Aggression subscale, mean ± SD*	34.4 ± 5.1
*AQ, Verbal Aggression subscale, mean ± SD*	13.3 ± 3.8
*AQ, Anger subscale, mean ± SD*	24.9 ± 5.0
*AQ, Hostility subscale, mean ± SD*	25.1 ± 6.6
**TAS-20 total score, mean ± SD**	53.6 ± 9.5
Non-alexithymia (range: ≤51), *n* (%)	572 (41.3%)
Possible alexithymia (range: 52–60), *n* (%)	496 (35.8%)
Alexithymia (range: ≥61), *n* (%)	317 (22.9%)
** *TAS-20, Difficulty Describing Feelings subscale, mean ± SD* **	14.5 ± 4.4
** *TAS-20, Difficulty Identifying Feelings subscale, mean ± SD* **	20.0 ± 4.9
** *TAS-20, Externally-Oriented Thinking subscale, mean ± SD* **	19.1 ± 4.3
**BSMAS total score, mean ± SD**	20.5 ± 6.6
Pathological level (range: ≥16), *n* (%)	1078 (77.8%)
**Binge Watching scale total score, mean ± SD**	20.2 ± 9.2

*n*: frequency; %: percentage; SD: standard deviation.

## Data Availability

The data presented in this study are available on request from the corresponding author.

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
