# Peer review of "COVID-19-Related Social Isolation Predispose to Problematic Internet and Online Video Gaming Use in Italy"

_ijerph, 2022, doi:10.3390/ijerph19031539_

Round 1

Reviewer 1 Report

The presented article has been comprehensively and competently supplemented and corrected by the authors. The structure of the article is logical, factual and substantively justified. In this regard, I would consider the possibility of including in the title of the information that the research project and the conclusions generated on its basis concern Italy. However, this is a loose suggestion, the inclusion of which would make the content of the article more precise in relation to the reader's expectations.
The adopted research methodology has been comprehensively and legibly described. From this point of view, the research process does not raise any doubts. The presentation of the research results is matter-of-fact, but overly elaborate tables and slightly complicated charts make it difficult for the reader to understand the authors' way of thinking. However, I don't know how to simplify these elements without losing the value of the content they represent. Thus, I admit the presentation techniques used as acceptable and justified.
The conclusion is clear, it clearly reflects the essence of the research and the results obtained. I think it would be advisable to have a clearer indication of how research can be continued, extended and how to demonstrate its attractiveness for comparative research.

Author Response

#Reviewer 1 Comments:

1) The presented article has been comprehensively and competently supplemented and corrected by the authors. The structure of the article is logical, factual and substantively justified. In this regard, I would consider the possibility of including in the title of the information that the research project and the conclusions generated on its basis concern Italy. However, this is a loose suggestion, the inclusion of which would make the content of the article more precise in relation to the reader's expectations.

We thank the reviewer for the positive feedback. According to the reviewer’s suggestion, we have properly modify the title, by clearly specifying the data here presented are referred to the Italian population.  

2) The adopted research methodology has been comprehensively and legibly described. From this point of view, the research process does not raise any doubts. The presentation of the research results is matter-of-fact, but overly elaborate tables and slightly complicated charts make it difficult for the reader to understand the authors' way of thinking. However, I don't know how to simplify these elements without losing the value of the content they represent. Thus, I admit the presentation techniques used as acceptable and justified.

We thank the reviewer for the positive feedback. We greatly appreciate your comment.

3) The conclusion is clear, it clearly reflects the essence of the research and the results obtained. I think it would be advisable to have a clearer indication of how research can be continued, extended and how to demonstrate its attractiveness for comparative research.

Dear reviewer, thank you for your suggestion and comment. Accordingly, we have properly modified the conclusion section, by clearly specifying the future research directions and research projects to be implemented to better deepen our findings.

Reviewer 2 Report

Dear authors,

Thank you so much for your hard work on this manuscript. I still have some questions about this manuscript.

  1. The authors used many sentences to descript the Statistical analysis on Page 5. But there was not a good order for the use of basic analysis (descriptive statistics), ANOVA, regression and further Medication analysis. Please reorder this part with more clearer method, eg. use subheads, or bullets/numberings.
  2. The Abbreviations of special terms have been highlighted in the contents. but I want to point the name of variables in the figures. Eg. in Figure 1, the IAT means Problematic internet use. Please unify their use.
  3. In the discussion, although the authors gave good discussions about related variables (internet usage, videgaming, etc), please focus on the terms used in the discussions. Because the same Abbreviations may be not used in the related articles/references. Please check them again.
  4. Could you give more p value of the Mediation effect in the anaylysis?

Author Response

#Reviewer 2 Comments:

Dear authors, Thank you so much for your hard work on this manuscript. I still have some questions about this manuscript. 1) The authors used many sentences to descript the Statistical analysis on Page 5. But there was not a good order for the use of basic analysis (descriptive statistics), ANOVA, regression and further Medication analysis. Please reorder this part with more clearer method, eg. use subheads, or bullets/numberings.

We thank the reviewer for the suggestion. Accordingly, we have properly modified the statistical section within the methodology heading, by clearly ordering all statistical tests used (see pages 5-6).  

2) The Abbreviations of special terms have been highlighted in the contents. but I want to point the name of variables in the figures. Eg. in Figure 1, the IAT means Problematic internet use. Please unify their use.

We thank the reviewer for the valuable suggestion. Accordingly, we have modified all captions of each figure in the text, by clearly specified the name of variables included (see Figure 1, Figure 2, Figure 3, Figure 4, Figure 5, Figure 6, Figure 7, and Figure 8).

3) In the discussion, although the authors gave good discussions about related variables (internet usage, videgaming, etc), please focus on the terms used in the discussions. Because the same Abbreviations may be not used in the related articles/references. Please check them again.

We thank the reviewer for the valuable recommendation. Accordingly, we have checked again the discussion section regarding the terms used and properly modified it (see pages 14-17 for more details).

4) Could you give more p value of the Mediation effect in the analysis?

We thank the reviewer for the suggestion. Accordingly, we have properly included p-values of the mediation effect (see throughout the manuscript).

This manuscript is a resubmission of an earlier submission. The following is a list of the peer review reports and author responses from that submission.

Round 1

Reviewer 1 Report

Thank you for the opportunity to review this paper. 

The paper reports on the extent of the Internet and videogame use during COVID-19 pandemic.  

I applaud to the authors for well conducted and up-to-date study and for brining up such important issue combining behavioral addictions and covid pandemic.

However, I see some major limitations of the paper.  The main one relates to sampling. The study was conducted as an online survey and thus represents only the population of online users and not the general population (also, more than half of the sample was between 18-29 years; vast majority was single, females are overrepresented (62.5%)).

Authors mention this in the limitations in the Discussion section. But the paper appears as national survey at a first sight.

Despite the above, the study is scientifically sound, well conduced.

My comments below are only some examples that I highlight. I have also provided feedback for each of the different sections of the paper for further consideration.

Abstract: 

-it should clearly state the period of data collection and country/region

-terminology; I recommend not to use the labeling term pathological video gaming but use, for example, at-risk or problematic video gaming instead.

Introduction 

-the first paragraph could be reduced if needed.

Materials and Methods

-1st paragraph: The list of institutions should not be part of the paper and could be listed as part, e.g., Acknowledgements. Also the institutions involved are listed as affiliations of the authors.

-in 2.1 Survey sampling the authors say that „The survey took approximately 15 minutes to be completed.“ Nevertheless, looking at the Clinical measures and the total number of items I doubt the survey was completed in 15 minutes. Or the authors should better explain what the 15 minutes relate to. 

Results

- I think that the number of figures should be reduced.

(Note, it would be interesting to conduct some sub-analysis on the difference between cohort below and above 30 years of age.)

Minor:

-the authors say: „The present study has been carried out as a nationwide population-based online survey for evaluating the impact of problematic Internet and videogames use…“ but I wonder about the sampling procedure and the size of the sample. Was the sample constructed as a national representative sample? Females were overrepresented (62.5%); was the sample stratified.

Author Response

#1 Reviewers' Comments:

1.1) Thank you for the opportunity to review this paper. The paper reports on the extent of the Internet and videogame use during COVID-19 pandemic. I applaud to the authors for well conducted and up-to-date study and for bringing up such important issue combining behavioral addictions and covid pandemic. However, I see some major limitations of the paper. The main one relates to sampling. The study was conducted as an online survey and thus represents only the population of online users and not the general population (also, more than half of the sample was between 18-29 years; vast majority was single, females are overrepresented (62.5%)). Authors mention this in the limitations in the Discussion section. But the paper appears as national survey at a first sight. Despite the above, the study is scientifically sound, well conduced. My comments below are only some examples that I highlight. I have also provided feedback for each of the different sections of the paper for further consideration.

We thank the reviewer for this comment and we do acknowledge the above-mentioned limitations, which have been now more thoroughly discussed in the discussion section (page 15, lines 26-52; page 16, lines 51-55, page 17, lines 1-4). For the sake of clarity, we’d like to point out that this was a nation-wide survey directed to the general adult population (which in Italy includes everyone above the age of 18 years), with a snowball sampling approach (a very common and widely used strategy during the pandemic emergency), so that we had no chance to choose or select respondents and thus to ensure a more stratified and inclusive selection approach. We also see that a significant part of our sample is represented by young subjects and, indeed, we plan to carry out a post-hoc specific analysis focusing specifically on this sub-sample only (i.e., excluding subjects older than 25 years of age, according to the WHO definition of “youth” people), in a separate paper.

1.2) Abstract: -it should clearly state the period of data collection and country/region; -terminology; I recommend not to use the labeling term pathological video gaming but use, for example, at-risk or problematic video gaming instead.

We thank the reviewer for this comment and valuable suggestion. Accordingly, we have properly revised the abstract, by including the period of data collection and country (Italy)(page 1). While regarding the Italian regions, data have been collected across all Italian regions. Moreover, the term ‘pathological’ videogaming has been accordingly replaced by the term ‘problematic’ videogaming (see throughout the manuscript).  

1.3) Introduction: the first paragraph could be reduced if needed.

We thank the reviewer for this advice. Accordingly, we have properly revised the Introduction section by shortening it (page 2).

1.4) Materials and Methods -1st paragraph: The list of institutions should not be part of the paper and could be listed as part, e.g., Acknowledgements. Also the institutions involved are listed as affiliations of the authors. -in 2.1 Survey sampling the authors say that „The survey took approximately 15 minutes to be completed.“ Nevertheless, looking at the Clinical measures and the total number of items I doubt the survey was completed in 15 minutes. Or the authors should better explain what the 15 minutes relate to. 

We thank the reviewer for this valuable suggestion. According to your comments, we have properly revised the manuscript as follows: a) we have shifted the list of institutions in the acknowledgment section (page 18, lines 15-19). We completely agree with your consideration regarding the time needed to complete the survey, it takes 45 minutes (we wrongly wrote the time for an editing mistake). For this reason, we have properly revised the materials and methods section accordingly (page 3, see paragraphs ‘2.1.Study design’ and ‘2.2. Survey Sampling’).

1.5) Results. I think that the number of figures should be reduced. (Note, it would be interesting to conduct some sub-analysis on the difference between cohort below and above 30 years of age).

We thank the reviewer for your comment. We preferred to include more figures in substitution of the text, as we collected a huge amount of data and to simplify the report of these findings, we decided to adopt a more graphical manner to report our results. Therefore, we hope that the reviewer may agree with our consideration. As we have already above stated, we also noticed that a significant part of our sample is represented by young subjects and, indeed, we plan to carry out a post-hoc specific analysis focusing specifically on this sub-sample only (i.e., excluding subjects older than 25 years of age, according to the WHO definition of “youth” people), in a separate paper.  

1.6) Minor. -the authors say: „The present study has been carried out as a nationwide population-based online survey for evaluating the impact of problematic Internet and videogames use…“ but I wonder about the sampling procedure and the size of the sample. Was the sample constructed as a national representative sample? Females were overrepresented (62.5%); was the sample stratified.

We thank the reviewer for this valuable comment. For the sake of clarity, we’d like to point out that this was a nation-wide survey directed to the general adult population (which in Italy includes everyone above the age of 18 years), with a snowball sampling approach (a very common and widely used strategy during the pandemic emergency), so that we had no chance to choose or select respondents and thus to ensure a more stratified and inclusive selection approach. However, according to reviewer’s suggestion, we have properly specified the section regarding the sampling procedure and national representativeness of the sample size needed for generalizing our findings (page 3, lines 27-41). Indeed, we also carried out a gender-based stratification of the results, all significant changes have been properly included in the main text (whereas not specified, there were not found statistical significant differences between male versus female sample). Moreover, we completely agree with the comment of the reviewer that our sample appears to be slightly overrepresented by female participants. This finding, even though enough representative of our national Italian general population (http://dati.istat.it/Index.aspx?QueryId=18460), appears slightly greater in percentage compared to our national data. Therefore, this issue has been properly underlined in our discussion section.

Reviewer 2 Report

Please refer to the file, thank you.

Author Response

#2 Reviewers' Comments:

2.1) The research issue of the study is interesting. However, the paragraphs’ description totally is redundant and lengthy, which made it a little difficult to read through and get the points.

We thank the reviewer for this comment and we have modified the manuscript accordingly, in particular by avoiding possible redundancy and shortening the text where appropriate (see page 2, Introduction paragraph; pages 14-17, Discussion section; page 17, Conclusion section).

2.2) Second, the measurements in this study are abundant, but some of the questions/items in the questionnaire/scales seem to be inherently interrelated or relevant to each other, like the “Internet Addiction Test” and the “Bergen Social Media Addiction Scale”. How to ensure the effectiveness of analysis methods.

We thank the reviewer for this observation and admit that clearly some of the above mentioned measures are inter-related. On the other hand, one might consider that also the phenomena that such psychometric instruments aim to measure are inter-related and it would not be a surprise if such indices could covariate. Indeed, we explored possible correlation and association by means of a wide array of analyses (including Pearson correlation, linear regression and mediation/moderation) which, in our opinion and in the absence of a clearer clue by this referee, may represent an effective way to analyze these data that would highlight possible overlaps and differences among the different psychopathological domains.

2.3) Last, both reliability test and validity test should be done to assess the reliability and validity of the indicators/questionnaire used in the study. All these above made this study need to make more improvements to meet the requirement for publication in the Journal of International Journal of Environmental Research and Public Health.

We thank the reviewer for recalling such a crucial issue as that of reliability/validity of the questionnaires used  in our study. Actually, all psychometric instruments used in our experimental work were officially validated and a list of each test with the related validation paper is provided below. However, we deemed not to describe all this information in detail in order not make the paper too lengthy and in the light of the consideration that anyone interested in such methodological aspects could have found this information in the scientific literature starting by reading the provided reference for each article. The only exception to the above consideration would be the questionnaire investigating binge watching behavior, which we declared overtly to be an ad hoc psychometric instrument, whose validation was not the aim of the present paper and it is probably not necessary, given the easy nature of the low number of items included in the questionnaire.

Questionnaire

Reference paper

Validity/reliability

Internet Addiction test (IAT) – Italian version

Fioravanti, G., Casale, S. Evaluation of the psychometric properties of the Italian Internet Addiction Test. Cyberpsychol Behav Soc Netw. 2015; 18(2): 120-128.

In order to explore the psychometric properties of the Italian version of the IAT, the original data set (n = 840) was randomly divided into two equal subsamples: one for EFA and the other for CFA. The EFA was conducted first to identify the underlying factor structure of the IAT scale. The CFA was then performed in order to validate the results of the EFA (…).

According to the KMO criterion, sampling adequacy was excellent (KMO = 0.91). Bartlett’s test of sphericity showed that the correlation matrix was suitable for factor analysis (v2 = 3,214.41, df = 190, p < 0.001). Using the conventional criterion for retaining factors with eigenvalues > 1.0 and the scree plot, a two-factor solution was identified, with the extracted factors explaining 45.59% of the total variance. All items loaded at 0.30 or above… Cronbach’s alpha values did not increase when an item was deleted, and all item-corrected total correlations were above 0.30…

After adding these constraints (due to their similar content), an acceptable fit for the two-factor solution was obtained (see Table 2). The path diagram and the standardized path coefficients are shown in Figure 1.

Correlations between the IAT total score and gender, age, online time in a typical week, and the GPIUS2 and its subscales were computed… IAT score correlated highly with the GPIUS2 total score and with the GPIUS2 subscales scores (Table 3).

Internet Gaming Disorder Scale –Short Form (IGDS9-SF) – Italian Version

Monacis, L., Palo, V., Griffiths, M.D., Sinatra, M. Validation of the Internet Gaming Disorder Scale - Short-Form (IGDS9-SF) in an Italian-speaking sample. J Behav Addict. 2016; 5(4):683-90

In order to explore the psychometric properties of the Italian version of the IGDS9-SF, the original data set (n=757) was recruited and confirmatory factor analysis and multi-group analyses were applied to assess the construct validity. Reliability analyses comprised the average variance extracted, the standard error of measurement and the factor determinacy coefficient. The convergent validity was assessed by correlating the IGDS9-SF scores with the scores of two similar scales (i.e., the GAS and the IAT), and the criterion validity was evaluated through patterns of correlations between the IGDS9-SF and the BSNAS scores. The BSNAS was chosen because it utilizes the same six behavioral addiction criteria used for the IGDS9-SF (…) Regarding the reliability analyses, once the single-factor solution was confirmed, the extent to which the items of the specific factor converged or shared a high proportion of variance was assessed through the VAE method. The result provided a good value (AVE = .76). In addition, the SEM was calculated to assess the degree to which the observed scores fluctuated as a result of the measurement errors. As expected, the value met the criterion (SEM = 1.79 ≤ SD/2 = 4.48). Finally, the factor score determinacy coefficient was .99, showing an excellent degree of internal consistency.

Depression, Anxiety and Stress Scale (DASS-21) – Italian Version

Bottesi, G., Ghisi, M., Altoe, G., Conforti, E., Melli, G., Sica, C. The Italian version of the Depression Anxiety Stress Scales-21: Factor structure and psychometric properties on community and clinical samples. Compr Psychiatry. 2015; 60:170-181.

In order to explore the psychometric properties of the Italian version of the DASS-21, the original data set of a community sample (n=417) and two clinical groups (32 depressive patients and 25 anxious patients) were recruited and a confirmatory factor analysis was carried out. The unidimensional model demonstrated the worse fit (χ 2 (189, n = 417) = 656.275, p < .001; NNFI = .901; CFI = .911; RMSEA = .077), whereas the three-factor oblique model showed good fit indices: (χ 2 (186, n = 417) = 353.672, p < .001; NNFI = .964; CFI = .968; RMSEA = .046). Correlations between factors in the three-factor oblique model were strong: anxiety-depression r = .69, anxiety-stress r = .74, and depression-stress r = .69. The bifactor model resulted the best factor solution, χ 2 (168, n = 417) = 271.292, p < .001; NNFI = .975; CFI = .980; RMSEA = .038. The ΔCFI between the bifactor and the three-factor oblique model was .012, thus supporting the hypothesis that the bifactor model is the most appropriate in reproducing the observed data. Cronbach's alpha coefficients exceeded .70 both in the community and clinical samples as a whole; no item removal was indicated to improve internal consistency values, thus indicating good to excellent internal consistency. Corrected item-total correlations were never smaller than .30 in either group in either scale, while mean inter-item correlations were always well above .40, which is considered an adequate value for narrow constructs [48] .

Correlations among the three subscales were overall medium, both for the community (mean r = .59, i.e., 35% of common variance) and clinical samples (mean for group: r = .54, i.e., 29% of common variance). Not surprisingly, in both groups the correlation between each DASS-21 subscale and the total score was large.

Lastly, two-week test–retest reliability values computed on the undergraduate student sample were large for all the DASS-21 scale scores (DASS-21 anxiety scale: r = .64; DASS-21 depression scale: r = .75; DASS-21 stress scale: r = .64; DASS-21 total: r = .74; all p s < .001).

Barratt Impulsiveness Scale –short form (BIS-15) – Italian Version

Spinella, M. Normative data and a short form of the Barratt Impulsiveness Scale. Int J Neurosci. 2007; 117(3):359-68.

Psychometric properties of the short Italian form of the BIS in a non-clinical, community sample constituted by 700 individuals recruited by word-of-mouth. The 30 items of the BIS were subjected to a factor analysis by principal components using Varimax rotation. A three factor solution was specified a priori based on previous studies showing a three-factor structure (Patton et al., 1995). Items factored in a manner that was highly consistent with the three established factors: nonplanning, motor impulsivity, and attention impulsivity. Five items with the highest loadings from each of the 3 factors were chosen and the factor analysis was re-run using only these 15 items. Eigenvalues, variance explained and intrascale reliability (Cronbach's alpha) for each item is reported in Table 3. Collectively, these three factors explained 33.4% of the variance. Cronbach's alpha for the total scale was very good (.81). As with the original scale, six of the items are inverted because they relate to lower impulsivity (e.g., “I plan for the future.”). Descriptive statistics for the 15 item BIS (BIS15) are also given in Table 1. A linear regression of demographic variables predicting the total score of the BIS was significant, F(3, 695) = 26.5, p < .001 (Table 2). The model accounted for 10.7% of the variance (Adjusted R2 = .103). Scores decreased with age and education. The sex differences (males scoring higher than females) fell to marginal significance (p = .059). However, the coefficients for all three demographic variables were equivalent. Scores of the short form were also normally distributed, as indicated by a Kolmogorov-Smirnov test (Z = 1.15, p = .14, two-tailed significance). BIS15 scores correlated with the total scores of the full test (BIS30) (r = .94, p < .001), and they also correlated with the total of the remainder items not included in the BIS15 (r = .65, p < .001).

Social Adaptation Self-evaluation Scale (SASS) – Italian Version

Bos, M., Dubini, A., Polin. V. Development and validation of a social functioning scale, the Social Adaptation Self-evaluation Scale. Eur Neuropsychopharmacol. 1997; 7:57-70.

The reliability of the instrument using Cronbach’s alpha is α = 0.752. The reliability of the following subscales was also assessed: job and spare time (α = 0.516); family and external relationship (α = 0.736); intellectual interest (α = 0.700); social compliance (α = 0.391) and control of surroundings (α = 0.326).

Buss-Perry Aggression Questionnaire (AQ) – Italian Version

Fossati A, Maffei C, Acquarii E, Di Ceglie A. Multigroup Confirmatory Component and Factor Analyses of the Italian Version of the Aggression Questionnaire. Eur J Psychological Assessment, 54-65, 2003.

In order to explore the psychometric properties of the Italian version of the AQ, the original data set of 392 university students and 563 high-school students were administered the original AQ by Buss and Perry (1992). Pooled correlation matrices of the AQ items, and pooled covariance matrices of the AQ scales were used in later confirmatory component and factor analyses, respectively. (…) The results of this study were in line with previous findings (Buss & Perry, 1992; Harris, 1995; Meesters et al., 1996) concerning the factor structure of the Aggression Questionnaire and confirmed in two independent samples of Italian students the 4-factor structure of the Aggression Questionnaire items described by Buss and Perry (1992). Quasi-inferential parallel analy- sis showed that four factors were necessary to explain the correlations among the Aggression Questionnaire items in both university and high-school student samples, con- firming Buss and Perry’s (1992) and Bernstein and Gesn’s (1997) findings (…) Scale-level analyses confirmed and extended the item-level factor analysis findings. Buss and Perry’s (1992) unidimensional model of the four AQ scales was the best-fitting model in both samples considered in this study. According to our results, not only the factor load- ings, but also the AQ scale intercepts and factor means were invalid, giving further evidence that aggression could be considered a personality trait temporally stable from adolescence to young adulthood. No substantial effect of subjects’ gender on the AQ scale factor structure was observed in this study (…)

Toronto Alexithymia Scale (TAS-20) - Italian Version

Bressi C, Taylor G, Parker J, Bressi S, Brambilla V, Aguglia E, Allegranti I, Bongiorno A, Giberti F, Bucca M, Todarello O, Callegari C, Vender S, Gala C, Invernizzi G. Cross validation of the factor structure of the 20-item Toronto Alexithymia Scale: an Italian multicenter study. J Psychosom Res. 1996 Dec;41(6):551-9. doi: 10.1016/s0022-3999(96)00228-0. 

The Italian translation of the TAS-20 has demonstrated factorial validity, internal consistency in normal adult and clinical samples, and high test–retest reliability over 2 weeks (…).

Bergen Social Media Addiction Scale (BSMAS) – Italian Version

Monacis, L., de Palo, V., Griffiths, M.D., Sinatra, M. Social networking addiction, attachment               style, and validation of the Italian version of the Bergen Social Media Addiction Scale. J Behav Addict. 2017; 6(2):178-186.

the scale reliability was examined using: (a) the average variance extracted (AVE) that assesses the extent to which the items of a specific factor converge or share a high proportion of variance (Hair, Black, Babin, & Anderson, 2010), values greater than 0.50 are considered adequate; (b) the standard error of measurement (SEM) that assesses the degree to which the observed scores fluctuate as a result of the measurement errors (Morrow, Jackson, Disch, & Mood, 2011). The criterion of acceptable precision was SEM ≤ SD/2 (Wuang, Su, & Huang, 2012); and (c) the factor determinacy coefficient of the internal consistency (Tabachnick & Fidell, 2013). As noted by Brown (2003), this coefficient represents an important result of factor analysis. In particular, a high degree of determinacy indicates that “the factor score estimates could serve as suitable substitutes for the factor itself” (Brown, 2003, p. 1418). Factor score determinacy represents the correlation between the estimated and true factor scores. It ranges from 0 to 1 and describes how well the factor is measured, with 1 being the best value (Muthén & Muthén, 1998–2012). The larger the coefficient (e.g., ≥0.70, Tabachnick & Fidell, 2013), the more stable the factors, in the sense that the observed variables account for substantial variance in the factor scores, whereas low values mean the factors are poorly defined by the observed variables (…)The internal consistency for the scale was examined using Cronbach’s α and was very good (α = .88). The inter-item correlations were relatively high (i.e., ≥0.30). The extent to which the items of the specific factor converged or shared a high proportion of variance was assessed through the AVE method. The results provided an adequate value (AVE = 0.57). The SEM was calculated to assess the degree to which the observed scores fluctuated as a result of the measurement errors. As expected, the value met the criterion (SEM = 2.02 ≤ SD/2 = 2.94). Finally, the factor score determinacy coefficient was 0.95, showing an excellent degree of internal consistency (…)

2.4) Abstract. When the abbreviation first appeared, it should be marked with the full name. The “Methods” you used need also to be clarified before you describe the results.

We thank the reviewer for this valuable suggestion. Accordingly, we have properly revised the abstract, also according to the comments of another reviewer (see page 1).

2.5) Introduction. 1. The Second paragraph and the Third, the “problematic Internet use” and its negative impact of language description needs to be considered to be re-framed. 2. I found that sentences sometimes involve more than two references, and the sentences were too long. The language needs to be concise.

We thank the reviewer for this comment. Accordingly, we have properly revised the second and the third paragraph of the introduction section. The introduction has been also shortened (see page 2). Moreover, the manuscript, including the Introduction section, has been English edited and accordingly revised (see throughout the manuscript).

2.6) Methods and Materials. 1. Is the first paragraph “study design”? If it is, please add the headline and describe a little bit more. 2. Please provide the reliability and validity analysis results of these scales in the 2.2 Clinical Measures part. 3. Again, some of the questions/items in the questionnaire/scales seem to be inherently interrelated or relevant to each other, please give more explanations.

We thank the reviewer for this comment. Accordingly, the materials and methods section has been properly revised. The first paragraph has been named ‘study design’ (page 3, lines 1-14).  The study design has been already extensively explained in the following publications of our research groups: reference 27 and reference 28. Moreover, as we already stated above, all psychometric instruments used in our experimental work were officially validated. However, we deemed not to describe all this information in detail in order not make the paper too lengthy and in the light of the consideration that anyone interested in such methodological aspects could have found this information in the scientific literature starting by reading the provided reference for each article. The only exception to the above consideration would be the questionnaire investigating binge watching behavior, which we declared overtly to be an ad hoc psychometric instrument, whose validation was not the aim of the present paper and it is probably not necessary, given the easy nature of the low number of items included in the questionnaire. In addition, we admit that clearly some of the above mentioned measures are inter-related. On the other hand, one might consider that also the phenomena that such psychometric instruments aim to measure are inter-related and it would not be a surprise if such indices could covariate. Indeed, we explored possible correlation and association by means of a wide array of analyses (including Pearson correlation, linear regression and mediation/moderation) which, in our opinion and in the absence of a clearer clue by this referee, may represent an effective way to analyze these data that would highlight possible overlaps and differences among the different psychopathological domains.

2.7) Results: 1. In Table 1, the Age group, why did you divide into 5 categories with different age intervals (18-19; 20-25; 30-39)? Please give more explanation. 2. The refer of “Table S1 and Table S2” was used too many times in the main part, which showed the two tables seemed to be important. Please consider how to do the Table display, or how to refer them better.

We thank the reviewer for this comment. The five categories with different age intervals have been used according to the I.Stat. (Italian National Institute of Statistics) modality to stratify epidemiological data related to the Italian general population. As you suggested, Table S1 and Table S2 have been properly included in the main text only once time (see page 5, line 32).

2.8) Discussions: 1. In Page 15. “Furthermore...”, the paragraph is too long. Please revise. 2. Regarding to the Limitations, after completing the description of the current problems, it is better to add the future research plan or solution to them.

We thank the reviewer for this suggestion. The discussion has been properly revised, also according to the suggestions provided by the other reviewers (page 15, lines 26-52; page 16, lines 51-55, page 17, lines 1-4). Many sentences have been shortened and revised. Regarding the future research plan or solution to them, we furtherly integrated the conclusion section by adding further considerations at this regard (page 17, lines 21-51).

2.9) References. Please standardize and unify the format of references.

We thank the reviewer for this suggestion. Accordingly, the reference section has been checked out and revised (see pages 18-21).

Reviewer 3 Report

Thank you for inviting me to review the article. I congratulate the authors on taking up an important, interesting and very timely topic.
The issue of media addiction (mainly internet addiction) during a pandemic is undertaken by many researchers representing various disciplines. In the peer-reviewed article, we are dealing with a well-prepared empirical project that provides interesting data. This makes it possible to confirm some of the known and intuitively generated hypotheses. I would like to draw the authors' attention to the following elements that could be improved:
- in terms of methodological description, it should be stipulated that the data obtained from the respondents are declarations - and therefore the results should be treated taking into account the inference errors caused by this factor
- the tables in the text are very extensive and extensive. I would suggest eliminating from them the data that the authors do not refer to in the text
- I would make a greater reservation that due to the specifics of the study, its conclusions cannot be fully extrapolated.
I also suggest verifying the language of the publication - sometimes it is very hermetic, which makes it impossible for a large group of readers to fully understand the intentions of the authors.

Author Response

#3 Reviewers' Comments:

3.1) Thank you for inviting me to review the article. I congratulate the authors on taking up an important, interesting and very timely topic.
The issue of media addiction (mainly internet addiction) during a pandemic is undertaken by many researchers representing various disciplines. In the peer-reviewed article, we are dealing with a well-prepared empirical project that provides interesting data. This makes it possible to confirm some of the known and intuitively generated hypotheses. 

We thank the reviewer for your positive feedback. Accordingly, we have properly revised our manuscript to adhere to your suggestions and valuable comments.

3.2) I would like to draw the authors' attention to the following elements that could be improved: - in terms of methodological description, it should be stipulated that the data obtained from the respondents are declarations - and therefore the results should be treated taking into account the inference errors caused by this factor.

We thank the reviewer for the valuable suggestion. Accordingly, we have revised the methodological section, by including that data obtained from respondents are declarations (page 3, lines 40-41) and we properly discussed this major limitation in the discussion section, by considering this potential inference error (page 16, lines 30-40).

3.3) - the tables in the text are very extensive and extensive. I would suggest eliminating from them the data that the authors do not refer to in the text. I would make a greater reservation that due to the specifics of the study, its conclusions cannot be fully extrapolated.

We thank the reviewer for this comment. Accordingly, we have checked out all redundant data already included in the Table 1 and Table 2 and we have revised the discussion section (page 15, lines 26-52; page 16, lines 51-55, page 17, lines 1-4).

3.4) I also suggest verifying the language of the publication - sometimes it is very hermetic, which makes it impossible for a large group of readers to fully understand the intentions of the authors.

We thank the reviewer for this advice. The manuscript has been thoroughly revised and checked for English editing (see throughout the manuscript).

Round 2

Reviewer 2 Report

It is an interesting study and I highly appreciate what the authors are trying to do. The manuscript is well done.

  1. However, the items' order in Table 2, and the Clinical Measures(page 3) described order is not matched orderly, which is not easy for reading.
  2. How many items are in BIS-15? please see Table 2, there were: range (16-54); and on Page 4, there were "15 items". How did you count the "16" and "54"?
  3. Where is the data (table display) about the IAT score in different populations? Please see the 3.2 part, Page 8-9. Like the IAT score of single respondents, nonprofessional gamers.
  4. I am not sure that whether the authors checked for the abbreviations in the whole text. Problematic Internet Use ==>IAT, Problematic Social Media Use ==>IDG9-SF, or IGDS-SF? The full names and abbreviations do not match in the whole manuscript.
  5. In Figures 2,3,4, the authors did the analysis all using DASS General, Depression, Anxiety, and Stress. Why did only use DASS General in Figure 1? Please give more explanation about these kinds of variables chosen in the analysis.

Author Response

To: Profs. Andrea Fiorillo

Maurizio Pompili

Gaia Sampogna

Ancona, October 28th, 2021

Subject: Cover letter - Resubmission of the paper ijerph-1395145 (title: “COVID-19-related Social Isolation predispose to Problematic Internet and Online Videogaming Use”).

Dear Editors,

We deeply thank you for giving us the possibility to revise our paper and for considering this revised version for publication in the IJERPH special issue on Mental Health in the Time of COVID-19.

   We did our best to properly address all referees’ comments and requests, which contributed significantly to improve the overall quality of the manuscript. For editors’ and referees’ ease, we report below here each comment of the first round of review, followed by our answer (marked in red color). We very much hope that our revised paper is now up to the quality standards of your respected journal and thus suitable for publication in IJERPH.

We thank you once again for reconsidering our study and please do not hesitate to contact us anytime, should you have any further doubt or query.

With kindest wishes,

Yours sincerely,

Umberto Volpe (on behalf of all co-authors)

Professor Umberto VolpeUnit of Clinical Psychiatry

Department of Neurosciences/DIMSC, School of Medicine

Polytechnic University of Marche, Via Conca 71, 60126, Ancona.

T: +39 71 596 3301

F: +39 71 596 3540

E-mail: u.volpe@staff.univpm.it

#2 Reviewers' Comments:

2.1) It is an interesting study and I highly appreciate what the authors are trying to do. The manuscript is well done.

We thank the reviewer for this positive feedback.

2.2) However, the items' order in Table 2, and the Clinical Measures (page 3) described order is not matched orderly, which is not easy for reading.

We thank the reviewer for this observation. Accordingly, we have modified the order of Table 2, according to the order of clinical measures described in paragraph 2.3 (see Page 7, Table 2).

2.3) How many items are in BIS-15? Please see Table 2, there were: range (16-54); and on Page 4, there were "15 items". How did you count the "16" and "54"?

We thank the reviewer for your comment. The BIS-15 as described in paragraph 2.3 Clinical Measures is the brief version of the BIS-11 tool which was originally constituted by 30 items. The BIS-15 is a 15-items, 4-point Likert-type scale containing both direct and reverse point items. The total score ranges from 15 to 60. We properly corrected the Table 2 (see page 7, Table 2). 

  • Where is the data (table display) about the IAT score in different populations? Please see the
  • Part, Page 8-9. Like the IAT score of single respondents, nonprofessional gamers.

We thank the reviewer for this observation. The Table 2 represents the IAT levels in the total sample according to the different severity ranges. The Table 2 did not report IAT total score in different populations as we summarized the findings in the main text, as already asked by previous reviewers. All detailed data are available to all readers upon request to the authors as repository file.   

2.5) I am not sure that whether the authors checked for the abbreviations in the whole text. Problematic Internet Use ==>IAT, Problematic Social Media Use ==>IDG9-SF, or IGDS-SF? The full names and abbreviations do not match in the whole manuscript.

We thank the reviewer for this observation. The abbreviation for Problematic Internet Use is PIU which is measured as moderate severity IAT total scores. IAT is a tool to measure the level of Internet usage and not the abbreviation for PIU. Similarly, Problematic Social Media Use (abbreviated as PSMU) is not IGDS-9-SF, which represents the tool used to measure the level of Internet Gaming Disorder (abbreviated as IGD). However, following reviewer’s suggestion, we have properly checked again throughout the manuscript all full names and abbreviations to be sure that all do match accordingly in the whole manuscript (see throughout the manuscript for further details). Furthermore, we have specified for each clinical measures, which are used to assess PIU and IA (see page 4, lines 9-10), that IGDS-9-SF is used to assess PVGD and IGD (see page 4, line 13), that BSMAS is used to assess PSMU (see page 5, line 9). 

2.6) In Figures 2, 3, 4, the authors did the analysis all using DASS General, Depression, Anxiety, and Stress. Why did only use DASS General in Figure 1? Please give more explanation about these kinds of variables chosen in the analysis.

We thank the reviewer for this observation. We have included only DASS Depressive subscale (and not DASS General total score) in Figure 1 as we reported in the figures only the clinically significant findings, as we already stated in the main text. In fact, regarding Figure 1, mediation analyses showed that IAT significantly predicts IGDS only by DASS depressive subscale and this association is not statistically significant mediated by DASS total score either by other DASS subscales. Conversely, for other mediation analyses described in Figures 2, 3 and 4, these associations have been demonstrated to be mediated significantly by all subscales and DASS total score, as illustrated in the main text and in each Figure 2, 3 and 4.